# In Pursuit of Healthy Aging: Effects of Nutrition on Brain Function

**DOI:** 10.3390/ijms22095026

**Published:** 2021-05-10

**Authors:** Thayza Martins Melzer, Luana Meller Manosso, Suk-yu Yau, Joana Gil-Mohapel, Patricia S. Brocardo

**Affiliations:** 1Neuroscience Graduate Program, Federal University of Santa Catarina, Florianópolis 88040-900, SC, Brazil; thayza.melzer@posgrad.ufsc.br; 2Translational Psychiatry Laboratory, Graduate Program in Health Sciences, University of Southern Santa Catarina, Criciúma 88806-000, SC, Brazil; luana.manosso@gmail.com; 3Department of Rehabilitation Sciences, Hong Kong Polytechnic University, Hung Hom, Hong Kong, China; 4Division of Medical Sciences, University of Victoria, Victoria, BC V8P 5C2, Canada; jgil@uvic.ca; 5Island Medical Program, Faculty of Medicine, University of British Columbia, Victoria, BC V8P 5C2, Canada

**Keywords:** aging, cognition, macronutrients, microbiota-gut-brain axis, micronutrients, neurodegeneration, nutrition

## Abstract

Consuming a balanced, nutritious diet is important for maintaining health, especially as individuals age. Several studies suggest that consuming a diet rich in antioxidants and anti-inflammatory components such as those found in fruits, nuts, vegetables, and fish may reduce age-related cognitive decline and the risk of developing various neurodegenerative diseases. Numerous studies have been published over the last decade focusing on nutrition and how this impacts health. The main objective of the current article is to review the data linking the role of diet and nutrition with aging and age-related cognitive decline. Specifically, we discuss the roles of micronutrients and macronutrients and provide an overview of how the gut microbiota-gut-brain axis and nutrition impact brain function in general and cognitive processes in particular during aging. We propose that dietary interventions designed to optimize the levels of macro and micronutrients and maximize the functioning of the microbiota-gut-brain axis can be of therapeutic value for improving cognitive functioning, particularly during aging.

## 1. Introduction

One of the most important factors that contribute to maintaining good health is consuming all the nutrients that the body needs to function properly [1]. According to the World Health Organization (WHO) guidelines, nutrition represents a critical part of health and development. Balanced nutrition is essential to physical and mental wellbeing throughout the entire lifespan, starting with early development during both the prenatal and postnatal stages, as well as all the way until the later stages of life [2].

The increase in average life expectancy is one of society’s outstanding achievements, and this has been associated with a shift in the leading causes of morbidity and mortality, from infectious diseases to non-communicable conditions [3,4] such as diabetes, cardiovascular disease, hypertension, and stroke. The WHO guidelines highlight that the correct intake of nutrients contributes to a robust immune system, lower risk of non-communicable diseases, and ultimately, an increase in longevity [2]. Numerous epidemiological studies have shown a correlation between optimum nutrition and a decrease in cancer incidence, the reversal of chronic low-grade systemic inflammation, chronic pain, and other chronic diseases such as rheumatoid arthritis [5,6,7,8]. Specific diets can slow down symptoms of these diseases. For example, several studies indicate that the long-term consumption of a Mediterranean diet (rich in fruits, vegetables, and olive oil) correlates with better cognition in aged populations [9,10]. It seems that the same is true for neurodegenerative diseases such as Parkinson’s disease (PD), Alzheimer’s disease (AD), and other types of dementia [11,12,13,14,15,16]. In these cases, the available evidence has suggested that nutrition could potentially modify the onset and trajectory of these diseases through changes in biochemical and epigenetic factors [17,18].

## 2. Cognition and Aging

The word “cognition” summarizes a series of processes that take place in our brain and that allow us to understand the world around us [19]. In other words, cognitive functions, such as attention, language, learning, memory, and perception, allow us to derive representations of relevant information from sensory inputs [20]. This information can be processed by the brain and used to modify behavior when necessary.

Cognitive processes are highly dependent on the prefrontal cortex, the brain area responsible for complex thinking, information analysis, and decision making [21], as well as the hippocampus and surrounding limbic structures, which are involved in learning and memory consolidation [22]. These processes are present in everyday life, at any age, in health and disease (Figure 1). In addition, for these processes to take place, the brain’s structure and function need to be intact and preserved.

Numerous intrinsic and extrinsic factors can contribute to the preservation of brain structure and function. These include, but are not limited to, intrinsic changes in neuronal plasticity and brain circuitry, exposure to different types of experiences and stimuli, physical activity, caloric, and nutritional intake, and age [23,24,25,26].

Indeed, aging in itself (i.e., independently from disease) is well known to be associated with significant changes in brain morphology, plasticity, and function [27,28]. In addition to this, aging in itself is one of the main risk factors for several diseases, especially neurodegenerative conditions. Moreover, age-associated changes in the brain’s structure, connectivity, and intrinsic metabolic pathways are thought to contribute, at least in part, to an increased susceptibility to neurodegenerative diseases, such as various types of dementia, AD, and PD, which have a high prevalence among the aging population [29]. Of note, most (if not all) neurodegenerative diseases are associated with various degrees of cognitive impairment, and the decline in cognitive function is often progressive, and, in many cases, culminates in profound dementia [30,31].

## 3. Nutrition

The more straightforward way of describing a well-balanced intake of nutrients is to portray a diversified diet where the individual can consume all the necessary macronutrients and micronutrients in appropriate proportions to maintain a working body. Such diet includes a variety of fruits and vegetables, legumes (e.g., lentils and beans), nuts, and whole grains (e.g., unprocessed maize, millet, oats, wheat, and brown rice). In addition, avoiding excess fats, sugar, salt, and processed goods is also highly recommended [32]. Such diet is believed to provide numerous health benefits and reduce the risk of several chronic conditions such as diabetes, hypertension, metabolic syndrome, etc. [33,34].

Carbohydrates, proteins, and fats are the macronutrients that act as primary energy sources (Figure 1). Carbohydrates are biomolecules formed primarily by carbon, hydrogen, and oxygen atoms. They can be monosaccharides (such as glucose, the primary source of energy for all cells), disaccharides or polysaccharides, depending on the size of the carbon chain [35]. In addition to being the main energy source, carbohydrates also act as molecular beacons and participate in the formation of nucleic acids [36].

Proteins are polymers of amino acids formed by carbon, oxygen, hydrogen, nitrogen, and sulfur [37]. These macronutrients perform several crucial functions in the body, including immune defense, cell communication, catalysis of reactions, transport of substances, and forming the supporting structure of the body.

Fats or lipids are composed of glycerol and fatty acids and can be divided into various categories, including phospholipids, glycerides, and steroids [38]. These molecules are also a source of energy, in which they are kept in the body as energy reserves and, in the absence of carbohydrates, can be metabolized to generate energy. Lipids form the lipid bilayer of cell membranes, are precursors for several hormones, and facilitate the transport of fat-soluble vitamins [39].

Micronutrients, such as vitamins and minerals (Figure 2), are needed in much smaller quantities than macronutrients but are essential for numerous metabolic, biochemical, and regulatory processes [40]. Vitamins can be divided into water-soluble and fat-soluble, depending on their solubility [41,42,43].

Of note, a balanced nutrition is not only linked to individual health. Individuals with adequate nutrition are less likely to develop numerous chronic conditions and therefore significantly decrease the burden that such conditions impose on health care systems.

### Nutrition and the Aging Process

The overall population continues to age. Data from 2019 shows that there were 703 million people over the age of 65 globally, and according to the United Nations estimates, the population over the age of 60 will reach 1.5 billion by 2050 [44]. On the other hand, the worldwide prevalence of dementia is around 50 million cases [45]. According to the World Alzheimer Report 2015, the chances an older adult has of developing some type of dementia is 2–4% at 65 years of age and increases to 15% at 80 years of age. As the population grows older, current estimates predict over 130 million cases by 2050 [46].

Given this, strategies that can promote healthy aging and offset the development of numerous chronic health conditions associated with aging are becoming increasingly relevant. Indeed, numerous studies have evaluated the potential beneficial effects of nutritional strategies in delaying the onset of age-related diseases and slowing down the progression of some conditions [47,48,49]. Several lines of evidence suggest that cognitive impairment and the risk of brain diseases associated with cognitive deficits including dementia can be lowered through the intake of specific macro and micronutrients present in balanced diets [50,51,52,53] and that several “special foods” can prevent or mitigate the degenerative processes associated with age, particularly B vitamins, flavonoids, and long-chain ω-3 fatty acids [54,55,56].

Indeed, some studies have shown that nutritional supplementation improves cognitive function in patients with AD. On a randomized, double-blind, placebo-controlled clinical trial, 210 AD patients were randomly divided into intervention (800 IU/day of vitamin D) and control groups (starch granules) for 12 months. The intervention group scored better in a cognitive assessment after 6 and 12 months of supplementation [57]. Similar evidence is found in other studies. These results, however, have methodological limitations regarding study subjects, vitamin dosage, duration of intervention, and individual nutritional history [58]. Nevertheless, evidence suggests that the most appropriate approach to maximizing the health benefits of nutrients is to consume a varied, multi-nutrient diet rather than supplementing particular nutrients [59,60]. In support, several studies have reported a significant decrease in the risk of developing AD with increased folate [61,62] or vitamin B-6 [61] consumption. In addition, a marginal reduction in the risk of AD and dementia has also been reported with increased total n-3 fatty acids, DHA, and fish consumption [54,56,63,64,65]. However, it is still currently unclear if/how nutrition can impact and modulate cognitive function once a neurodegenerative process is initiated in the brain.

## 4. Microbiota-Gut-Brain Axis and Nutrition

The reciprocal impact of the gastrointestinal (GI) tract on brain function has been recognized since the 19th century. Presently, the microbiota-gut-brain axis is believed to form a bidirectional homeostatic communication pathway, through which the GI tract exerts an influence on brain function, and vice versa [66,67,68,69]. Indeed, all GI functions (including, motility, secretion, mucosal maintenance, and immunological defense) require regulation and coordination provided by the enteric nervous system (ENS) [70]. It is estimated that the human microbiota contains around 10^14^ bacterial cells, a number that is ten times greater than the number of human cells [71]. The microbiota composition is host-specific and can change according to age, sex, medication, diet, exercise, and multiple other factors [72,73].

An emerging body of evidence supports the role of the microbiota-gut-brain axis on several neurological disorders, including neurodevelopmental disorders (for example, autism spectrum disorder), psychiatric disorders (for example, major depressive disorder, anxiety, and schizophrenia), and neurodegenerative disorders (for example, AD and PD) [74,75,76,77,78]. Indeed, clinical evidence has shown that gut microbiota changes in patients afflicted with these disorders [74,79,80,81,82].

This microbiota-gut-brain axis bidirectional communication occurs through many pathways, as outlined in the following paragraphs:

Neuroanatomic communication: The main neuroanatomical communication between the ENS (which directly innervates the GI tract) and the CNS is provided by the vagus nerve (parasympathetic input) and spinal nerves (sympathetic input). It is worth mentioning that, although the communication is bidirectional, almost 90% of vagal fibers are afferent, suggesting that the brain is primarily a receiver of information, rather than a transmitter, with regards to gut-brain communication [70]. The afferent fibers of the vagus nerve reach the nucleus tractus solitarius in the brain stem and, from there, send information to other regions of the CNS, including regions related to cognition. The afferent fibers associated with the spinal nerves that provide innervation to the GI tract reach the CNS mainly via the spinothalamic tract, synapsing at the thalamus, and from there, information is sent to other brain regions [83,84]. Interestingly, the vagus nerve has receptors for some hormones and neurotransmitters that are produced in the GI tract, such as serotonin, cholecystokinin (CCK), YY peptide (PYY), and ghrelin, as well as receptors for bacterial fragments, such as lipopolysaccharides (LPS). Thus, the activation of these receptors can signal the brain regarding what is happening in the gut. In addition, short-chain fatty acids produced by the gut microbiota can also activate afferent fibers of the vagus nerve [84,85]. Along these lines and further supporting the microbiota-gut-brain communication, it has recently been shown that vagotomy prevents the beneficial neurobehavioral effects induced by probiotics [86]. In addition, clinical studies also suggest a beneficial role for vagus nerve stimulation in cognition [87].

Hormones and neurotransmitters: The GI tract produces more than 30 hormones and signaling molecules that influence many physiological processes and act on various tissues. At the CNS level, these hormones affect the brain centers that regulate appetite, metabolic control, and behavioral pathways linked to reward, mood, anxiety, stress, memory, etc. Some of these molecules can enter the systemic circulation, cross the blood-brain barrier (BBB), and reach the CNS. Alternatively, many can also act locally and activate afferent vagal terminals in the gut, thus generating afferent signals [88,89]. Through enteroendocrine cells, enteric neurons, and gut microbiota, the GI tract can also synthesize neurotransmitters that influence the functioning of both the GI tract and the CNS [90,91]. The neurotransmitters produced by the GI tract and gut microbiota can directly interact with the enteric and peripheral nervous systems or communicate with the brain via stimulation of vagal nerve innervation or the immune system, which in turn may influence the CNS. Of note, serotonin, catecholamines (dopamine, epinephrine, and norepinephrine), acetylcholine, and gamma-aminobutyric acid (GABA) (neurotransmitters known to impact several brain functions, including cognition [92,93,94,95]) can all be synthesized in the GI tract [96,97].

Neuroendocrine pathways: Another factor that influences microbiota-gut-brain communication is stress. In the face of acute stress, the body responds by activating the hypothalamic-pituitary-adrenal (HPA) axis, which will lead to a release of cortisol by the adrenal gland. When the HPA axis is functioning normally, the neuroendocrine stress response is counter-regulated by a negative feedback mechanism. However, in some situations, such as in the case of persistent stress and/or exaggerated intensity, a deleterious effect on HPA axis regulation can be potentially detrimental to the body due to excess cortisol [98,99,100]. Stress can cause several GI tract changes, including changes in GI motility, GI secretions, and intestinal permeability, while also causing adverse effects on the gut microbiota [101]. Conversely, changes in microbiota can also alter the stress response. A study with germ-free animals has shown that these animals have exaggerated HPA stress response compared to animals colonized with beneficial bacteria [102]. In line with this, another preclinical study demonstrated that microbiota can modulate stress-dependent activation of pituitary and adrenal glands [103]. Noteworthy, at the CNS level, it is well recognized that chronic stress can affect the brain (and in particular hippocampal) functioning, including learning and memory processes, as well as mood regulation [99,104,105].

Immune system/inflammation: The GI tract contains about 70–80% of the body’s immune cells [106]. The intestinal microbiota plays an essential role in the development and modulation of the intestinal immune system. Components of Gram-negative bacteria such as lipopolysaccharides (LPS) can interact with Toll-like receptors (TLR), influencing the immune response and the production of inflammatory cytokines, such as interleukins (IL) [107,108]. Peripherally produced cytokines can reach or send signals to the CNS through several mechanisms and result in neuroinflammation, which can affect the brain function and contribute to the pathophysiology of various neurological diseases [109,110,111]. On the other hand, the CNS regulates innate immune responses through neuronal and hormonal routes [112].

Bacteria-derived metabolites: Metabolites from bacteria, such as short-chain fatty acids (SCFAs), including acetate, propionate, and butyrate, can also influence the gut microbiota-gut-brain crosstalk [89]. SCFAs are essential metabolites produced by bacterial fermentation of substrate (mostly dietary fiber) in the large intestine. Although the functions of SCFAs are not yet completely elucidated, it is believed that they may have local effects in the colon (e.g., decrease inflammation, improve mucus production), affect gene expression by inhibiting histone deacetylases, impact hormone regulation (e.g., glucagon-like peptide 1 and peptide YY), and interact with vagal afferents. Moreover, SCFAs can also regulate BBB integrity/function and cross the BBB, acting directly on the brain [113,114].

Neurotrophic factors: Brain-derived neurotrophic factor (BDNF) is arguably one of the best characterized CNS neurotrophic factors. It is a key molecule involved in synaptic and structural plasticity, learning, and memory. Moreover, BDNF is associated with the pathophysiology and treatment of several neurological diseases [115,116]. Interestingly, intestinal microbiota can play a role in regulating BDNF levels in the CNS. Indeed, a preclinical study revealed that germ-free mice (mice that had no exposure to microorganisms) had reduced BDNF expression in the cerebral cortex and hippocampus compared to controls [102]. Another study also demonstrated that germ-free mice had lower BDNF mRNA expression in the hippocampus, cingulate cortex, and amygdala [117]. Moreover, it has been shown that peripheral administration of LPS also decreases BDNF levels in the brain [118]. On the other hand, animals supplemented with probiotics or prebiotics showed increased BDNF levels in various brain regions [119,120].

Numerous intrinsic and extrinsic factors have been shown to influence the microbiota-gut-brain axis. These include genetic and epigenetic factors, as well as several environmental factors, such as exercise, exposure/consumption of different drugs, and consumption of probiotics (including their mode of delivery) [72]. Moreover, diet composition and nutritional status have repeatedly been shown to be one of the most critical modifiable factors regulating the gut microbiota at different time points across the lifespan and under various health conditions [121].

Within this scenario, the type of diet is a major factor that can affect the gut microbiota and the gut-brain crosstalk. The western diet (characterized by a high consumption of refined, processed foods, saturated fat, trans fat, and sugars, as well as a low intake of fruits and vegetables) has been shown to alter the gut microbiota and the functioning of the GI tract, alter the formation of SCFAs, and increase inflammation, all of which can, in turn, impact the microbiota-gut-brain axis [121,122]. In addition, most industrialized foods contain food additives and advanced glycation end products, which can also alter gut microbiota, increase inflammation, and affect the permeability of the GI tract [123,124]. On the other hand, a Mediterranean diet (characterized by a high consumption of fruits, vegetables, whole grains, fiber, nuts, legumes, and olive oil, as well as a low consumption of sugars, saturated fat, red and processed meats, and industrialized foods) has been shown to improve gut microbiota, reduce inflammation, and ensure the adequate production of hormones, neurotransmitters, and bacteria-derived metabolites [121,122]. In addition, a healthy diet provides a source of dietary polyphenols, a broad group of secondary plant metabolites that can modulate the microbiota-gut-brain connection by acting both at the intestine and the brain levels, since some polyphenols can cross the BBB [125].

It is important to highlight that the gut microbiota can also synthesize vitamins and influence the activation of polyphenols [126]. Some of the key nutrients that are important for gut health and, consequently, can influence the microbiota-gut-brain axis, are summarized below:

Omega-3 fatty acids: Modulate gut microbiota composition and maintain gut immunity/inflammation [127,128].

Vitamin D: Modulates the immune response of the intestine as well as the gut microbiota. Induces the expression of several antimicrobial peptides in dendritic cells and contributes to maintaining adequate tight junction formation [129,130,131].

Vitamin A: Modulates the immune response of the intestine as well as the gut microbiota. Involved in the differentiation of mucosal epithelial cells [132,133,134].

Vitamin E: Protects the intestinal barrier function and modulates the gut microbiota [135].

Zinc: Modulates the immune response of the intestine, the gut microbiota, and the integrity of the mucous membranes [136,137].

Iron: Critical for the replication and survival of most bacteria [138]. However, an excess of iron can impact gut microbiota composition [139].

## 5. Metabolic and Molecular Mechanisms That Contribute to Brain Aging and NeuroDegeneration

Aging biology is intimately associated with dysregulated metabolism, which is one of the hallmarks of aging. Metabolomics use analytical profiling techniques for measuring and comparing large numbers of metabolites present in organisms [140,141]. The use of metabolomics can provide a quantitative profile of metabolites altered with age [142,143,144]. Metabolomics studies of aging have found that the metabolic profiles are strongly correlated with chronological age [145,146,147,148] and have pointed to some hub metabolites (nicotinamide adenine dinucleotide, reduced nicotinamide dinucleotide phosphate, 〈-ketoglutarate, and ®-hydroxybutyrate) that appear to play a critical role in the metabolism and signaling pathways to control aging [149].

Neurons are subjected to metabolic, ionic, and oxidative stress due to their electrochemical activity, bioenergetics, and various types of cellular stressors such as oxidative stress and intracellular Ca^2+^ dysregulation [29]. Signaling pathways evolved to respond to cellular stresses adaptively and to ease immediate threat, alert other cells, and form defenses against future stressors [150,151]. However, while exposure to cellular stressors increases with age, the activity and efficacy of adaptive stress responses are thought to become increasingly impaired, rendering the CNS vulnerable to injury and neurodegenerative processes [152]. This section focuses on the changes in cell metabolism that occur during the aging process, including how dysfunctional organelles and the accumulation of certain metabolites can impact the aging brain.

Mitochondria are arguably one of the most important intracellular organelles. They are responsible for energy production through the synthesis of ATP, intracellular Ca^2+^ homeostasis, and regulation of nuclear gene transcription [153,154,155]. Mitochondria are also involved in apoptosis or programmed cell death, which occurs normally in the healthy brain, but when in excess, can be linked to either acute injury or chronic neurodegenerative processes [156].

Numerous age-related mitochondrial alterations have been reported, including morphological changes [157,158], increased oxidative damage to its mitochondrial DNA [159,160], impaired functioning of the electron transport chain (ETC) [161,162,163,164], and impaired Ca^2+^ handling [162,165]. In addition, and as part of the natural aging process, most brain cells accumulate dysfunctional mitochondria [166,167].

Another common feature of aging is an oxidative imbalance, i.e., an increase in the production of reactive oxygen species (ROS) and/or a reduction in antioxidant defenses. The major ROS that is produced in neurons is superoxide (O_2_^−^), which is generated during mitochondrial respiration [168]. Nitric oxide (NO), a reactive nitrogen species (RNS) involved in neuronal signaling, has been linked to vascular dysfunction in the aging cerebral cortex when produced in excess [169]. Peroxynitrite (ONOO^−^, which results from the combination of NO and O_2_^−^) and hydroxyl radical (HO) can initiate lipid peroxidation cascades that result in the damage of cell membranes [170]. Moreover, accumulation of 4-hydroxynonenal (HNE), a lipid peroxidation product, has been shown to be associated with the formation of amyloid deposits and neurofibrillary tangles in AD [171]. In addition, both HNE and NO can oxidize cysteine, lysine, and histidine, as well as tyrosine residues, resulting in an impaired protein function [170,172,173]. Finally, ROS accumulation can also cause DNA oxidation and the impairment of DNA repair mechanisms (through oxidation of protein enzymes involved in DNA repair) [174]. Indeed, both mitochondrial and nuclear DNA are regularly damaged by ROS during the normal processes of cellular metabolism. In neurons, oxidative damage to DNA is particularly increased following excitatory synaptic activity [175,176]. In healthy cells, oxidized DNA bases are readily removed and replaced by undamaged bases by protein enzymes involved in DNA repair pathways [177]. However, several studies have shown that aging is associated with an increase in the amount of oxidized mitochondrial and nuclear DNA and by a reduction in the activity of DNA repair mechanisms [178]. Indeed, it has been suggested that impaired DNA repair alone is sufficient to cause aging-like phenotypes, as evidenced in certain genetic diseases such as Cockayne syndrome [179].

In addition to mitochondrial changes and increased oxidative stress, intracellular Ca^2+^ homeostasis is also compromised in the aging brain. Studies performed on hippocampal pyramidal neurons demonstrated that aging results in an increased Ca^2+^ influx through Ca^2+^ channels, thus leading to an increase in intracellular Ca^2+^ concentration [180,181,182,183]. This dysregulation in intracellular Ca^2+^ levels can in turn impact protein phosphorylation, cytoskeletal dynamics, gene expression, mitochondrial function, and neurotransmitter release. Together, these intracellular changes can result in excitotoxicity and ultimately cell death [184].

Aging has also been associated with a decrease in the expression of several neurotrophic factors, including BDNF, nerve growth factor (NGF), and insulin growth factor 1 (IGF-1) signaling [185,186,187,188]. In turn, these deficits can also contribute to impaired neuronal mitochondrial function, intracellular Ca^2+^ handling, and antioxidant defenses during aging [185].

Of note, the cell has intrinsic mechanisms to counteract the damage induced by oxidative stress and/or intracellular Ca^2+^ dysregulation. Oxidized proteins are tagged for proteasomal degradation through the process of ubiquitination, while oxidized/damaged membranes, mitochondria, and other intracellular organelles are targeted for degradation by lysosomes through the process of autophagy [189,190,191]. However, excessive ROS/RNS production and a consequent increase in levels of oxidative stress can quickly overwhelm both the proteasomal and lysosomal degradation systems, rendering them ineffective and causing cell damage and demise [192,193,194]. In support of this, intracellular accumulation of autophagosomes with undegraded cargos, dysfunctional mitochondria, and polyubiquitinated proteins (indicating proteasomal dysfunction) have been found in elderly individuals [189,195,196] and are thought to eventually result in cell death [194]. Of note, various neuronal populations are thought to present different degrees of vulnerability to proteasome and lysosome dysfunction [197]. This differential vulnerability might contribute, at least in part, to the differences in neuronal susceptibility associated with various neurodegenerative diseases.

Other extrinsic factors are also known to contribute to neuronal degeneration in aging and/or neurodegenerative conditions. For example, it is well established that chronic exposure to uncontrolled stress (physical or psychological) can impair neuronal plasticity and predispose neurons to degeneration through hyperactivation of the hypothalamic-pituitary-adrenal (HPA) axis and the resulting increase in glucocorticoid levels [198].

Glucose concentration is another important extrinsic factor that can mediate age-related changes in the brain. Circulating glucose concentrations tend to rise during aging, as a result of impaired glucose transport in response to insulin [199,200]. In agreement, positron emission tomography (PET) imaging has shown insulin resistance and impaired glucose transport in neurons of elderly individuals, particularly in the temporal, parietal, and frontal (including motor) lobes [201]. Of note, this impairment in glucose transport seems to be more common in elderly subjects with mild cognitive impairment and AD when compared to control subjects [202]. In addition to insulin resistance, the neuronal glucose transporter (GLUT3), is vulnerable to HNE and can be easily oxidized [170,203]. On the other hand, the use of ketone bodies such as β-hydroxybutyrate (BHB) and acetoacetate do not appear to be affected by aging or neurodegeneration, as no changes in the levels of these metabolites have been observed in the brains of AD patients [204].

Together, all of the above age-induced cellular alterations culminate in neuronal dysfunction, and impaired functional (i.e., synaptic) and structural neuroplasticity (including adult hippocampal neurogenesis) [205,206,207,208,209]. Indeed, in the aged brain progenitor cells exhibit several features associated with mitochondrial dysregulation, including reduced mitochondrial oxidative metabolism [210], genetic alterations that affect the functioning of the mitochondrial ETC [211], and oxidative stress. In addition, impaired DNA repair, and inflammation can also contribute to the age-related reduction in adult neurogenesis seen with aging [212,213,214,215].

Finally, an increase in inflammation is also seen in the aging brain and thought to further contribute to neuronal dysfunction. In line with this, glial cells often exhibit an activated phenotype in the aged brain, leading to the release of proinflammatory cytokines, and tumor necrosis factor α (TNF-α) [216,217]. In addition, there is also evidence suggesting that the complement cascade may also be activated during aging, and this process may also contribute to the pathogenesis of chronic and acute degenerative processes, such as AD and ischemic stroke, respectively [218,219,220].

## 6. Prevention of Cognitive Decline through Nutritional Interventions

This section will review some of the nutritional interventions thought to have beneficial effects with regards to cognitive function.

A meta-analysis encompassing 15 trials and 6480 participants demonstrated that diet interventions (primarily based on the adoption of a Mediterranean diet) improved performance on measures of global cognition, executive function, and processing speed during normal aging [221]. This is also supported by a systematic review of the evidence, which indicates that adherence to a Mediterranean diet is associated with better cognitive performance. However, it should be noted that the majority of findings come from epidemiologic studies that provide evidence for a correlation between Mediterranean diet and cognition, and therefore, no cause-and-effect relationship can be asserted at this time [222].

The effects of other healthy diets on cognition and neurodegeneration have also been studied, including dietary approaches designed to prevent hypertension (DASH) [223,224], a Mediterranean-DASH diet Intervention for Neurological Delay (MIND) [224,225,226], and plant-based diets [227,228,229]. Although there are some differences among these diet patterns, it is worth noting that all of them are based on high consumption of fruits, vegetables, and whole grains and low consumption of sugars, saturated fat, and processed foods, which provide a high intake of polyphenols. The mechanisms associated with the benefits of these diets on cognition are varied, and include antioxidant and anti-inflammatory actions, modulation of gut microbiota, gut functions, and insulin sensitivity, increased neurotrophic support, and decreased neuronal damage [122,230,231].

Interestingly, these diets seem to have a long-term effect on the brain. A multicenter longitudinal study demonstrated that a Mediterranean diet and “A Priori Diet Quality Score” (APDQS) were associated with better midlife cognitive performance [232]. In addition, consumption of a wide variety of vegetables has been shown to be associated with a lower risk of global cognitive decline in older individuals [233].

A randomized clinical trial with 447 volunteers with normal cognitive function compared the effects of three distinct dietary interventions: A Mediterranean diet supplemented with extra-virgin olive oil (1 L/week), a Mediterranean diet supplemented with mixed nuts (30 g/d) or a control diet (advice to reduce dietary fat). Cognitive tests performed approximately 4 years after the intervention showed that a Mediterranean diet supplemented with olive oil or mixed nuts was associated with improved cognitive function [16].

In a similar study, 31 older adults with mild cognitive impairment were randomly assigned to ingestion of Brazil nuts (one unit per day) or no nut supplementation for a period of 6 months. Twenty participants completed the trial, and improvements in verbal fluency and constructional praxis were significantly increased in the nut-supplemented group when compared with the control group. Moreover, an increase in the serum levels of selenium and in the activity of erythrocyte glutathione peroxidase were detected in the Brazil nut-supplemented group [234]. Other clinical studies also support the idea that nut consumption is associated with increased cognitive performance [235,236,237], and this seems to be related to their antioxidant and anti-inflammatory properties, as well as anti-amyloidogenic effects [238].

Olive oil is the primary fat source in the Mediterranean diet and the benefits of this diet on brain function have been primarily attributed to it. Olive oil contains oleic acid (a mono-unsaturated fatty acid) and biophenols, which have multiple pharmacological actions, including antioxidant, anti-inflammatory, and anti-amyloidogenic properties. Together, these are thought to contribute to the neuroprotective properties of olive oil [239,240]. In agreement, a recent clinical study revealed a significant positive correlation between cognitive function and consumption of monounsaturated fatty acids and oleic acid in elderly Japanese individuals [241].

Fish intake also seems to provide benefits with regards to brain function. A meta-analysis found a borderline significant decrease (by 36%) in the risk to develop AD in individuals with a high intake of fish when compared with those with a low intake [242]. Another study suggested that moderate fish consumption during childhood and adolescence may be associated with some cognitive benefits. It was proposed that the consumption of fish during these developmental stages may potentially influence neuropsychological performance later on in adulthood [243]. Fish is a source of omega-3 polyunsaturated fatty acids, which are thought to be critical for brain functioning. Indeed, omega-3 fatty acids are thought to contribute to cell membrane integrity and fluidity, regulation of peripheral immune function, microglial activation, and brain glucose metabolism, as well as to increase BDNF levels and decrease amyloid-β (Aβ) load [244,245,246,247,248].

Cocoa flavonoids or dark chocolate (≥70% cocoa) intake also contribute to cognitive performance. Several clinical studies demonstrated that cocoa intake can improve neurovascular coupling, increase regional glucose metabolism in the occipital and visual cortex, and enhance normal cognitive functioning [249,250,251,252]. Moreover, a retrospective cohort study that included 55 patients diagnosed with amnestic mild cognitive impairment reported that cocoa flavonoids appear to decrease the progression of mild cognitive impairment to dementia [253]. In addition to neurovascular improvement, clinical and/or preclinical studies suggest that the benefits of cocoa ingestion on brain functioning may involve antioxidant and anti-inflammatory properties, increased BDNF levels, decreased levels of Aβ and cortisol, as well as improvement in cholinergic neurotransmission [254,255,256].

Interestingly, caffeine is another compound that has been shown to have potentially beneficial effects with regards to brain function. Indeed, a clinical trial with 2513 participants aged 60 years or older suggested that caffeine was associated with increased cognitive performance [257]. A prospective cohort study in older men also suggested that coffee consumption may also be associated with reduced cognitive decline [258]. Moreover, a coffee intake of ≥2 cups/day was shown to be significantly associated with lower Aβ accumulation when compared with a coffee intake of <2 cups/day, even after controlling for potential confounders [259]. As recently reviewed, coffee and its components may exert multiple effects on the digestive tract, including antioxidant and anti-inflammatory actions, which in turn can influence the health and functioning of the gut-brain axis [260].

Another drink that seems to benefit cognition is green tea (*Camellia sinensis*). A systematic review (that included three cohort studies and five cross-sectional studies) supported the hypothesis that green tea intake might reduce the risk for mild cognitive impairment and cognitive impairment associated with dementia and AD. It is believed that green tea and its components (polyphenols, caffeine, and L-theanine) can exert anti-inflammatory, antioxidant, and other neuroprotective actions [261,262].

Several micronutrients appear to be important for cognition. A clinical study reported an improvement in cognition following micronutrient supplementation (A-Z multivitamin/mineral supplement) in healthy participants [263]. On the other hand, a meta-analysis that compared plasma levels of micronutrients in AD patients demonstrated that these individuals had significantly lower plasma levels of folate, vitamin A, vitamin B12, vitamin C, and vitamin E when compared with healthy controls [264]. Many micronutrients play essential roles in brain functioning, being important for cellular energy production, cell maintenance, and repair, neurotransmitter synthesis, oxidation-reduction homeostasis, etc. [265]. Table 1 summarizes the clinical evidence supporting the role of micronutrients on cognition as well as their proposed mechanisms of action.

Conversely to the beneficial effects associated with healthy diets, it is worth noting that some eating habits can have a deleterious impact on cognition. These include a lower vegetable intake, non-adherence to dietary guidelines, increased intake of trans-fat as well as processed meat, sugar, and ultra-processed foods, as well as increased alcohol consumption [300,301,302]. Moreover, glycemic variability has also been shown to result in oxidative stress and neuroinflammation, ultimately contributing to cognitive dysfunction [303]. Furthermore, excess levels of some minerals such as copper, manganese, iron, and zinc seem to increase the risk and/or worsen the progression of AD [304,305,306]. Finally, another nutritional aspect that can impact cognition is obesity. A systematic review and meta-analysis suggested a positive association between obesity in mid-life and later dementia [307]. In line with this, another meta-analysis reported that obesity increased the risk for AD [308] and that obese individuals have broad executive function deficits [309].

In addition, it is worth noting that combining a healthy diet with other interventions are known to have beneficial effects (e.g., physical exercise) and will likely result in enhanced benefits and positive outcomes. In support, findings from a double-blind, randomized controlled trial with individuals aged 60–77 years suggest that a multidomain intervention (including diet, exercise, cognitive training, and vascular risk monitoring) can improve (or at least maintain) cognitive functioning in at-risk elderly people in the general population [51].

Finally, another important nutrition-related factor that is thought to influence aging is the timing of food ingestion in relation to intrinsic circadian rhythms. Indeed, circadian rhythms optimize physiology and health by temporally coordinating cellular function, tissue function, and behavior. However, these endogenous rhythms are less efficient with age, making feeding-fasting patterns an external cue that can potentiate daily biological rhythms. It is now well recognized that both intermittent and periodic fasting as well as time-restricted feeding (TRF, in which food consumption is restricted to certain hours of the day), can be quite beneficial and help prevent and/or ameliorate several diseases. In support, sustaining a robust feeding-fasting cycle, even without altering nutrition quality or quantity, can prevent or reverse various chronic diseases in experimental models. On the other hand, dysregulated eating patterns can disrupt the temporal coordination of metabolism and physiology leading to chronic diseases that are also characteristic of aging (for review see [310,311]).

## 7. Conclusions

While it is recognized that a dietary supply of macronutrients is essential for human health, micronutrients such as vitamins and trace minerals are also crucial for maintaining brain health. In contrast, deficiency or disturbances in any of these can be associated with brain dysfunction and contribute, at least in part, to the pathophysiology of several neurological disorders. On the other hand, interventions designed to optimize the levels of such macro and micronutrients can potentially form the basis for therapeutic strategies aimed at improving cognitive functioning, particularly in the aging brain (Figure 3). Moreover, diet and nutritional status are the most critical modifiable factors regulating the gut microbiota at different time points across the lifespan and under various health conditions. Therefore, advances in our understanding of the mechanisms underlying the actions of macro and micronutrients on the brain and the microbiota-gut-brain axis will facilitate the development of nutritional interventions aimed at optimizing brain function and preventing or treating neurodegenerative disorders and other age-related conditions.

## Figures and Tables

**Figure 1 ijms-22-05026-f001:**
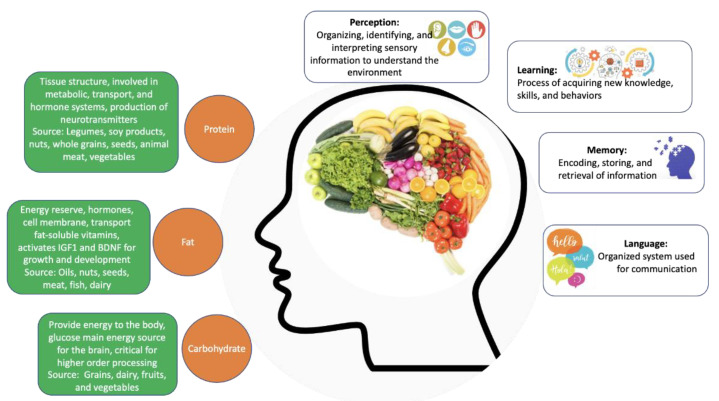
The contribution of individual macronutrients to brain health and cognitive function. Source: Author’s own work.

**Figure 2 ijms-22-05026-f002:**
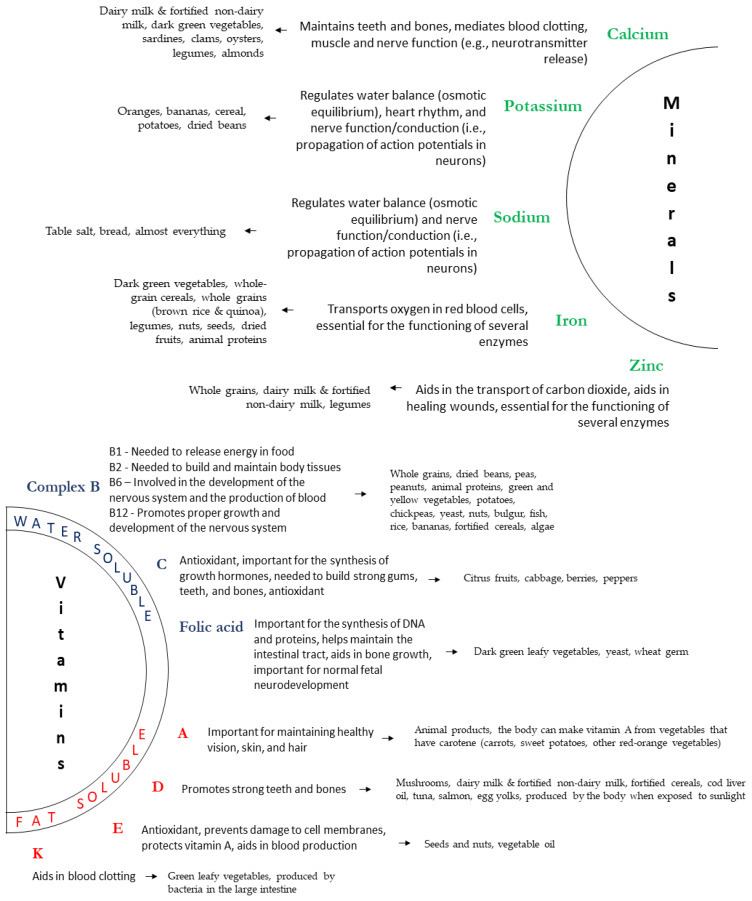
Main micronutrients and their sources and functions. Source: Author’s own work.

**Figure 3 ijms-22-05026-f003:**
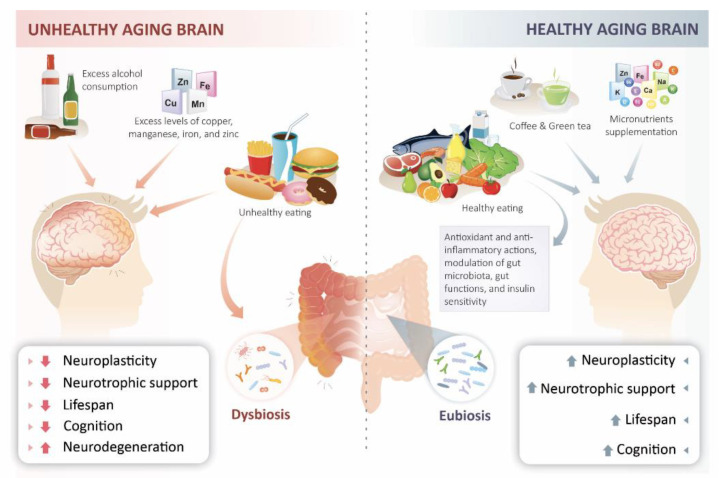
The contribution of diets to brain health during aging. Source: Author’s own work.

**Table 1 ijms-22-05026-t001:** The role of micronutrients on cognition and their potential mechanisms of action.

Vitamin/Mineral	Clinical Evidence	Possible Mechanisms of Action
B vitamins	Thiamine (vitamin B1) deficiency has been associated with cognitive impairment [266]. Dietary intake of vitamin B6 (pyridoxine) has been associated with better cognitive function [267]. B12 deficiency was shown to impair memory, and serum levels below 300 pmol/L were shown to cause irreversible hippocampus changes [268]. Folic acid (vitamin B9) supplementation was shown to significantly improve cognitive function [269]. Controversial data regarding the role of vitamins B6, B12, and folic acid on cognition have also been reported [270].	B vitamins act as co-enzymes for several catabolic and anabolic enzymatic reactions [271]. They can regulate the levels of homocysteine, and S-adenosylmethionine. They have anti-inflammatory [272] and antioxidant [273] properties.
Vitamin A	Increased cognitive decline was shown to be positively correlated with lower vitamin A levels and marginal vitamin A deficiency was shown to facilitate AD pathogenesis [274]. Vitamin A deficiency can be a predictor of mild cognitive impairment [275]. A higher intake of total carotenoids (which can be converted to vitamin A in the body) was shown to be associated with a decreased risk of moderate or poor cognitive function [276].	Marginal vitamin A deficiency starting in the embryonic period is thought to alter genes associated with AD [277]. Vitamin A can be converted to retinoic acid in the brain, which is essential for synaptic plasticity in regions of the brain involved in learning and memory, such as the hippocampus [278].
Vitamin K	Increased dietary vitamin K intake was to shown to be associated with better cognition in older adults [279,280]. Oral anticoagulants that are non-vitamin K antagonists were shown to be associated with a lower risk of cognitive impairment when compared with vitamin K antagonists or acetylsalicylic acid [281].	Vitamin K is involved in the ɤ-carboxylation of two vitamin K-dependent proteins whose activity contributes to adequate cerebral homeostasis: Gas-6 and protein S. Vitamin K participates as a co-factor in the synthesis of sphingolipids, which are essential constituents of cell membranes [282].
Vitamin D	Maintaining adequate vitamin D status during aging may contribute to a reduction in cognitive decline and a delay in the onset of dementia [283]. Low vitamin D levels were shown to be associated with worse cognitive performance and cognitive decline [284]. Vitamin D deficiency is thought to be a risk factor for AD [285]. The effects of vitamin D supplementation on improving cognition are still controversial [284].	Vitamin D contributes to cerebral activity in both the embryonic and adult brain [286]. Vitamin D regulates calcium homeostasis, clears Aβ peptide deposits, has antioxidant and anti-inflammatory effects, regulates brain plasticity, and improves neurogenesis [287,288,289].
Vitamins C and E	A decrease in mild cognitive impairment was observed in individuals with high plasma vitamin C concentrations [290]. Blood vitamin C concentration was shown to be significantly lower in individuals with dementia when compared with healthy controls [291]. Higher vitamin E levels were shown to be associated with higher scores on verbal memory, immediate recall, and better language/verbal fluency performance, particularly among a younger age group [292]. Controversial data regarding the role of vitamin E on cognition have also been reported [293].	Vitamins C and E are two important exogenous antioxidant molecules, which can decrease oxidative stress, neuroinflammation, and Aβ load [294,295,296].
Selenium	Circulating and brain selenium concentration was shown to be significantly lower in AD patients when compared to healthy controls [297,298].	Selenium has antioxidant properties. Selenoproteins regulate some neurotransmitters, including acetylcholine [299].

## Data Availability

Not applicable.

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
