# Peer review of "In Pursuit of Healthy Aging: Effects of Nutrition on Brain Function"

_ijms, 2021, doi:10.3390/ijms22095026_

Round 1

Reviewer 1 Report

Manuscript by Melzer T et al is an interesting Review on the role of Nutrition on brain aging.

Points of criticism:

  1. Findings from the large, long-term, randomised controlled Trial  showed that a multidomain Intervention (diet, exercise, cognitive Training) could improve or maintain cognitive functioning in at-risk elderly people from the general Population-FINGER Study. This data shoulb be introduced in the manuscript
  2. Authors should better describe the role of metabolomics in aging of the brain
  3. Authors should discuss the role of circadian rhythms, time-restricted feeding brain aging.
  4. Is it possible to introduce atleast on figure summarizing the effect of Nutrition on brain aging ??

Circadian rhythms, time-restricted feeding, and healthy aging.

Importance of cardiomedtabolic health as a major determinant of age related cognition function

Author Response

Findings from the large, long-term, randomised controlled Trial  showed that a multidomain Intervention (diet, exercise, cognitive Training) could improve or maintain cognitive functioning in at-risk elderly people from the general Population-FINGER Study. This edata should be introduced in the manuscript

Response: Please note that the FINGER-study has now been added to the revised review. The following paragraph summarizing the findings of this study was added at the end of Section 6: In addition, it is worth noting that combining a healthy diet with other interventions known to have beneficial effects (e.g., physical exercise) will likely result in enhanced benefits and positive outcomes. In support, findings from a double-blind, randomized controlled trial with individuals aged 60–77 years suggest that a multidomain intervention (including diet, exercise, cognitive training, and vascular risk monitoring) can improve or (or at least maintain) cognitive functioning in at-risk elderly people in the general population (Ngandu et al 2015).

Authors should better describe the role of metabolomics in aging of the brain

Response: Please see that the following description of the role of metabolomics in the aging brain was added to Section 5:

Aging biology is intimately associated with dysregulated metabolism, which is one of the hallmarks of aging. Metabolomics uses analytical profiling techniques to measure and compare large numbers of metabolites present in organisms (140,141). The use of metabolomics can provide a quantitative profile of metabolites altered with age (142-144). Metabolomics studies in aging have found that metabolic profiles are strongly correlated with chronological age (145-148) and have pointed to some hub metabolites (e.g., nicotinamide adenine dinucleotide, reduced nicotinamide dinucleotide phosphate, ?-ketoglutarate, and ?-hydroxybutyrate) that appear to play a critical role in the metabolism and signaling pathways during aging (149).

Authors should discuss the role of circadian rhythms, time-restricted feeding brain aging.

Response: We agree with the reviewer that circadian rhythms and their relationship with time-restricted feeding are also important to healthy aging. However, a detailed discussion on these topics is outside of the scope of the present review. Nevertheless, we have added the following paragraph at the end of Section 6 to highlight the importance of these factors for healthy aging:

Finally, another important nutrition-related factor that is thought to influence aging is the timing of food ingestion in relation to intrinsic circadian rhythms. Indeed, circadian rhythms optimize physiology and health by temporally coordinating cellular function, tissue function, and behavior. However, these endogenous rhythms are less efficient with age, making feeding-fasting patterns an external cue that can potentiate daily biological rhythms. It is now well recognized that both intermittent and periodic fasting as well as time-restricted feeding (TRF, in which food consumption is restricted to certain hours of the day), can be quite beneficial and help prevent and/or ameliorate several diseases. In support, sustaining a robust feeding-fasting cycle, even without altering nutrition quality or quantity, can prevent or reverse various chronic diseases in experimental models. On the other hand, dysregulated eating patterns can disrupt the temporal coordination of metabolism and physiology leading to chronic diseases that are also characteristic of aging [310; 311].

Is it possible to introduce at least one figure summarizing the effect of Nutrition on brain aging ??

Response: We agree with the reviewer's suggestion, and have now included a summary figure to summarize the main effects of nutrition on the aging brain.

Reviewer 2 Report

“In Pursuit of Healthy Aging: Effects of Nutrition on Brain Function”

Overall strengths of the article:

This review provides an overview of how nutrition impacts brain health and function, focusing on the linkage between nutrition and cognitive processes, particularly in the aging brain. The correct intake of nutrients contributes to a robust immune system, lower risk of non-communicable diseases, and ultimately, an increase in longevity. Numerous studies showed that specific diets can slow down symptoms of many chronic diseases. Overall this review is very well structured, organized, and interesting to read.

I did not see any critical weakness, though some formatting required.

Author Response

This review provides an overview of how nutrition impacts brain health and function, focusing on the linkage between nutrition and cognitive processes, particularly in the aging brain. The correct intake of nutrients contributes to a robust immune system, lower risk of non-communicable diseases, and ultimately, an increase in longevity. Numerous studies showed that specific diets can slow down symptoms of many chronic diseases. Overall this review is very well structured, organized, and interesting to read.

I did not see any critical weakness, though some formatting required.

Response: thank you so much for your kind comments

Reviewer 3 Report

Dear authors,

I thank you for you interesting manuscript. Please find my comment here under:

1) This is a review article: please explain and define your prisma search strategy to find relevant literature.

2) Line 44: you say: It seems that the same is true for neurodegenerative dis-42 eases such as Parkinson's disease (PD), Alzheimer's disease (AD), and other types of dementia [11–16]= both references 11 and 16 are review article. Please refer to specific articles which can support your statement.

3) Line 46: you say: In these cases, the available evidence has suggested that nutrition could potentially modify the onset and trajectory of these diseases through changes in biochemical and epigenetic factors [17]= the same as comment 1: please refer to specific article and not to a review article.

4) Line 62: you say: Numerous intrinsic and extrinsic factors can contribute to the preservation of brain structure and function. These include, but are not limited to, intrinsic changes in neuronal plasticity and brain circuitry, exposure to different types of experiences and stimuli, physical activity, caloric and nutritional intake, and age [22–27]. The same as comments before: please refer to specific article and not a review article.

5) Line 66: you say: Indeed, aging in itself (i.e., independently from disease) is well known to be associated with significant changes in brain morphology, plasticity, and function [27]. This reference is a book. You refer to a chapter with 20 pages. Please be more specific, chapter, page.

6) You say: Such diet is believed to provide numerous health benefits and reduce the risk of several chronic conditions such as diabetes, hypertension, metabolic syndrome, and others [32]. You refer to a table from a review article. Please refer, if possible, to a specific source.

7) Table 2: please report; source: adapted from ……

8) Line 128: You say: several "special foods" can prevent or mitigate the degenerative processes associated with age, particularly B vitamins, flavonoids, and long-chain ω-3 fatty acids [45,52–54] : reference 45 is a review article. Please specify you reference article, otherwise you have enough with references 52-54.

9) Table 3: please report; source adapted from …..

10) Line 138: reference number 57: you did not complete this item in your reference list.

11) line 145: It is very interesting paragraph but would you please be more specific about reference 65, 66 and 68.  you refer now to a lot of pages.

12) Line 271: you say: Omega-3 fatty acids: modulate gut microbiota composition and maintain gut immun-271 ity / inflammation [120,121]: Reference 121 is a review article. Please specify your source article.

13) Line 273-282: regarding reference 122: four each item (vit D, A, zinc) please specify your source article.

Author Response

1) This is a review article: please explain and define your prisma search strategy to find relevant literature.

Response: Since our review is a descriptive review and not a systematic review, we did not use the PRISMA strategy to search for relevant literature. However, some of the keywords used during our literature search included: aging AND food; diet; gut-brain-axis; microbiota-gut-brain axis; nutrients; supplementation; cognition; mediterranean diet.

2) Line 44: you say: It seems that the same is true for neurodegenerative diseases such as Parkinson's disease (PD), Alzheimer's disease (AD), and other types of dementia [11–16]= both references 11 and 16 are review articles. Please refer to specific articles which can support your statement.

Response: Please note that original articles are now cited to support this sentence.

3) Line 46: you say: In these cases, the available evidence has suggested that nutrition could potentially modify the onset and trajectory of these diseases through changes in biochemical and epigenetic factors [17]. the same as comment 1: please refer to specific article and not to a review article.

Response: Please note that original articles are now cited to support this sentence.

4) Line 62: you say: Numerous intrinsic and extrinsic factors can contribute to the preservation of brain structure and function. These include, but are not limited to, intrinsic changes in neuronal plasticity and brain circuitry, exposure to different types of experiences and stimuli, physical activity, caloric and nutritional intake, and age [22–27]. The same as comments before: please refer to specific article and not a review article.

Response: Please note that original articles are now cited to support this paragraph.

5) Line 66: you say: Indeed, aging in itself (i.e., independently from disease) is well known to be associated with significant changes in brain morphology, plasticity, and function [27]. This reference is a book. You refer to a chapter with 20 pages. Please be more specific, chapter, page.

Response:

6) You say: Such diet is believed to provide numerous health benefits and reduce the risk of several chronic conditions such as diabetes, hypertension, metabolic syndrome, and others [32]. You refer to a table from a review article. Please refer, if possible, to a specific source.

Response: Please note that original articles are now cited to support this sentence.

7) Table 2: please report; source: adapted from ……

Response: Please note that Table 2 has been substituted by the new Figure 1.

8) Line 128: You say: several "special foods" can prevent or mitigate the degenerative processes associated with age, particularly B vitamins, flavonoids, and long-chain ω-3 fatty acids [45,52–54] : reference 45 is a review article. Please specify you reference article, otherwise you have enough with references 52-54.

Response: Please note that original articles are now cited to support this sentence.

9) Table 3: please report; source adapted from …..

Response: Please note that now table 3 has now been modified as per the Reviewer’s suggestion.

10) Line 138: reference number 57: you did not complete this item in your reference list.

Response: Thank you for your comment. The citation of Reference #57 has now been fixed in the Reference List

11) line 145: It is very interesting paragraph but would you please be more specific about reference 65, 66 and 68.  you refer now to a lot of pages.

Response: Please note that original articles have now been cited to support this paragraph. .

 12) Line 271: you say: Omega-3 fatty acids: modulate gut microbiota composition and maintain gut immunity / inflammation [120,121]: Reference 121 is a review article. Please specify your source article.

Response: Please note that original articles are now cited to support this sentence.

13) Line 273-282: regarding reference 122: four each item (vit D, A, zinc) please specify your source article.

Response: Please note that now original articles are now cited to support this sentence.

Reviewer 4 Report

  1. In a review article, a pictorial or graphical representation is more appealing than the text part. In order to improve the quality of the article, the authors should include a few figures.
  2. The abstract is short; however, it should be revised briefly by outlining the need for this review, its purpose, and its importance.
  3. There is a lack of connectivity between the abstract and the conclusion.
  4. Tables 1 and 2 are very simple and not particularly focused on the subject.
  5. What does the term "unnecessary stimuli" entail in Table 1?
  6. overall the low is missing 

Author Response

In a review article, a pictorial or graphical representation is more appealing than the text part. In order to improve the quality of the article, the authors should include a few figures.

Response: We agree with the Reviewer and have now added 3 Figures to the manuscript.

The abstract is short; however, it should be revised briefly by outlining the need for this review, its purpose, and its importance.

Response: Please note that the purpose of the present review has now been clearly stated in the Abstract.

There is a lack of connectivity between the abstract and the conclusion.

Response: In response to the Reviewer’s comment, we have now edited the Abstract and the Conclusion, to better align the take-home messages of this review article in both sections.

Tables 1 and 2 are very simple and not particularly focused on the subject.

Response: Please note that Tables 1 and 2 have now been replaced by the new Figure 1.

What does the term "unnecessary stimuli" entail in Table 1?

Response: This term is no longer used, as Table 1 and Table 2 have now been replaced by the new Figure 1.